# Comparison of Acute Irisin and Cognitive Responses to Different Exercise Modalities Among Late Adolescents

**DOI:** 10.3390/healthcare13243242

**Published:** 2025-12-10

**Authors:** Yakup Zühtü Birinci, Serkan Pancar

**Affiliations:** 1Faculty of Sport Sciences, Bursa Uludağ University, Bursa 16059, Türkiye; 2Faculty of Sport Sciences, Aksaray University, Aksaray 68100, Türkiye; serkanpancar@aksaray.edu.tr

**Keywords:** BDNF, executive function, exercise, high-intensity interval training, irisin, lactate, neurotrophin

## Abstract

**Background/Objectives**: Exercise supports physical and cognitive health through neurotrophin-mediated pathways, with irisin playing a key role in neuroprotection and synaptic plasticity. As adolescence represents a period of heightened neuroplasticity and metabolic adaptation, determining how different exercise modalities influence neurotrophic and cognitive responses is critical for health promotion in youth. This study aimed to compare the acute effects of low-intensity continuous training (LICT), short-interval high-intensity interval training (SI-HIIT), and long-interval HIIT (LI-HIIT) on circulating irisin levels and executive function in healthy late adolescent males. **Methods**: Eleven participants completed all conditions in a randomized crossover design with a 7-day washout. Venous blood samples and the Trail-Making Test, Parts A and B (TMT-A, TMT-B) were assessed pre- and postexercise, with continuous heart rate monitoring. **Results**: Post-exercise irisin levels were significantly greater in both HIIT protocols (SI-HIIT, *p* < 0.001; LI-HIIT, *p* < 0.038) than in the LICT protocol. Only the SI-HIIT group presented significantly shorter TMT-A (vs. LICT, *p* < 0.001; vs. LI-HIIT, *p* = 0.016) and TMT-B (vs. LICT, *p* < 0.001; vs. LI-HIIT, *p* < 0.001) completion times post-exercise. **Conclusions**: A single HIIT session elicited greater increases in circulating irisin and executive function compared with LICT. These findings highlight exercise intensity and interval structure as key factors for enhancing neurocognitive health, offering valuable insight for developing early-life training strategies to promote brain health.

## 1. Introduction

Exercise is widely recognized as a form of non-pharmacological ‘medicine’, conferring broad systemic benefits across cardiovascular, metabolic, musculoskeletal, and neurobiological systems and is recommended from childhood through the later stages of life [1]. Despite these well-documented effects, global participation levels remain critically low [2]. Furthermore, adolescence may coincide with a decline in habitual physical activity, along with the early emergence of sedentary behavior patterns and increased susceptibility to metabolic dysregulation [3].

Against this backdrop, high-intensity interval training (HIIT) has gained prominence as a time-efficient training strategy that often produces superior adaptations compared with traditional low- or moderate-intensity continuous training (LICT, MICT). Notably, growing evidence indicates that HIIT induces greater improvements in cardiorespiratory fitness and maximal oxygen uptake (VO_2_max), even in children and adolescents, and may additionally confer benefits for cardiometabolic health [4,5,6]. These adaptations also appear to be influenced by the length and structure of the intervals, as shorter and longer bouts may differentially stimulate central and peripheral components of aerobic capacity [7]. HIIT further induces unique hemodynamic and molecular stimuli due to its intermittent structure, characterized by alternating bouts of exertion and recovery. These repeated fluctuations in blood flow and shear stress activate endothelial mechanotransduction pathways, promoting vascular remodeling and potentially enhancing cerebral perfusion [8]. In parallel, HIIT elicits pronounced redox-sensitive signaling cascades that contribute to both peripheral and central benefits, including improved metabolic regulation and neuroprotection [8]. These hemodynamic and redox stimuli are increasingly thought to be mediated by exercise-induced myokines, which act as molecular transducers converting mechanical and metabolic signals within skeletal muscle into systemic and brain-specific adaptations [9].

Among these myokines, irisin has attracted particular attention as a promising molecular mediator. It was first identified in 2012 by Boström et al. [10] as a cleaved product of the transmembrane precursor fibronectin type III domain-containing protein 5 (FNDC5) in skeletal muscle and is regulated by peroxisome proliferator-activated receptor γ coactivator-1α (PGC-1α), a transcriptional coactivator induced in response to muscle contraction. Initially, discovered for its role in promoting the browning of white adipose tissue and enhancing systemic energy expenditure, irisin has since been recognized as an exercise-induced myokine with pleiotropic effects on human health, including neurodegenerative [11], metabolic [12], and cardiovascular [13] diseases. There is growing interest in its potential role in muscle–brain crosstalk [14], particularly in mediating neuroprotective and cognitive-enhancing effects. Emerging evidence suggests that irisin can cross the blood–brain barrier and modulate neuroplasticity, notably by upregulating brain-derived neurotrophic factor (BDNF) expression in the hippocampus, thereby influencing learning, memory, and executive functions [15,16]. The relative contribution of these pathways to the exercise-induced augmentation of cerebral BDNF remains complex, with converging evidence from both animal and human studies underscoring the need for further mechanistic investigations into the role of irisin in neurocognitive adaptation.

Experimental models have demonstrated that the cognitive benefits of physical exercise critically depend on irisin signaling; deletion of FNDC5/irisin impairs exercise-induced enhancements in learning, memory, and hippocampal BDNF. Moreover, exogenous irisin administration restores these functions in models of aging, Alzheimer’s disease, physical inactivity, and cerebral ischemia by reducing neuronal apoptosis and promoting neuroplasticity [17,18]. Collectively, these findings indicate that irisin may act as a key molecular mediator linking skeletal muscle activity to brain function under both physiological and pathophysiological conditions. These preclinical findings provide strong evidence for the involvement of irisin in neurocognitive processes.

In addition to substantial preclinical evidence, human studies support these findings. Nevertheless, knowledge regarding the most effective exercise modality for eliciting optimal irisin responses remains limited, as existing research reports heterogeneous outcomes. A recent systematic review and meta-analysis demonstrated that continuous endurance training significantly increases circulating irisin in healthy adults, whereas evidence for the effects of interval training remains insufficient to draw definitive conclusions [19]. In contrast, Tsuchiya et al. [20] reported that high-intensity running in healthy sedentary men induced a sustained elevation in circulating irisin, whereas low-intensity running resulted in a reduction, indicating that exercise intensity can modulate irisin secretion independently of energy expenditure. In support of these results, Colpitts et al. [21] demonstrated that, in youth with a healthy weight, HIIT elicits a markedly greater peak irisin response than MICT does, whereas no such difference was observed in their overweight or obese peers. Similarly, Archundia-Herrera et al. [22] reported that, in sedentary overweight or obese female adolescents, a single bout of HIIT acutely increased irisin levels in skeletal muscle without affecting plasma concentrations, whereas MICT did not alter irisin levels in either muscle or plasma. However, although both studies indicate a superior effect of HIIT on irisin responses, the populations examined differ substantially. Löffler et al. [23] further demonstrated that short bouts of intensive exercise induced an acute increase (~1.2-fold) in serum irisin in both children and adults, whereas prolonged or chronic physical activity did not produce significant changes. Collectively, these heterogeneous findings may reflect the influence of individual characteristics [24], metabolic status (e.g., adiposity, muscle mass) [25], fitness level [26], and exercise modality [21,27] while being largely unaffected by daily fluctuations or meal intake [23].

While previous studies have provided valuable insights into the regulation of irisin under different exercise conditions, heterogeneous and often population-specific findings leave several key questions unanswered. Given this variability, focusing on adolescence becomes particularly relevant, as this developmental stage is characterized by ongoing maturation of the prefrontal cortex, synaptic reorganization, and the continued refinement of executive functions [28,29]. This period constitutes a unique neurodevelopmental milieu marked by heightened neuroplasticity and increased sensitivity to external stimuli such as exercise [30,31]. Exercise in adolescents has been shown to upregulate neurotrophic factors and is associated with improvements in cognitive functioning [32]. Furthermore, emerging evidence suggests that adolescents may exhibit distinct physiological reactivity to exercise compared with adults, although direct comparative evidence remains limited [33,34]. Therefore, adolescence may represent a biologically sensitive window in which to examine how different exercise modalities influence myokine signaling and cognition. Building on this rationale, investigating the modality-specific effects of exercise on irisin release and executive function in this population may provide valuable insights for early-life neuroprotection and the development of age-specific exercise strategies aimed at supporting long-term brain health.

Therefore, the present study aimed to compare the acute effects of LICT, short-interval high-intensity interval training (SI-HIIT), and long-interval high-intensity interval training (LI-HIIT) on circulating irisin and cognitive performance in healthy late adolescent males. It is hypothesized that, compared with LICT, HIIT protocols, particularly short-interval HIIT, would elicit greater increases in circulating irisin and enhanced cognitive performance.

## 2. Materials and Methods

### 2.1. Subjects

Sample size estimation was conducted via G*Power software (version 3.1.9.7; University of Düsseldorf, Düsseldorf, Germany) via ANOVA: repeated-measures, within/between interaction. The effect size was calculated on the basis of irisin levels between pre- and post-exercise in two groups, with a partial eta squared of 0.31 [35]. This corresponds to an effect size (f) of 0.67. The analysis was performed with an alpha level of 0.05 and a statistical power of 0.80 (1−β = 0.80), assuming 3 groups and 2 measurement points. The output parameters indicated a critical F value of 5.143 and a total required sample size of 9. A total of 16 individuals were screened, of whom 12 met the inclusion criteria and were randomized.

Although 12 participants were initially randomized, 1 discontinued the study due to an injury during the first intervention session. Consequently, eleven healthy late-adolescent males [age: 18.27 ± 0.46 years, height: 175.91 ± 3.01 cm, weight: 71.18 ± 3.57 kg; body mass index: 22.99 ± 0.67 kg/m^2^; fat percentage of body: 8.09 ± 1.3; maximum aerobic speed (MAS): 16.27 ± 0.98 km/h] completed all sessions and were included in the analyses.

Participants were recruited via social media announcements and printed posters on university campuses and local sports facilities. Eligibility screening required being a native speaker with normal or corrected-to-normal vision, possessing valid medical clearance, and having no serious lower-limb injury within the past 12 months or recent use of performance-enhancing medications or supplements. Individuals were excluded if they reported smoking, alcohol or drug use, psychoactive substance use, or any history of depression, neurological disorders, or neuromotor/musculoskeletal impairments. All eligible participants were informed about the procedures and provided written informed consent before each session.

The study design and reporting followed the Consolidated Standards of Reporting Trials (CONSORT) guidelines and the Standard Protocol Items: Recommendations for Interventional Trials (SPIRIT). Specific CONSORT extensions, including those for within-subject designs [36] and nonpharmacological interventions [37] (see Figure 1), were also considered. Given the within-subject and nonpharmacological nature of the design, blinding was not possible; therefore, both the participants and researchers were aware of the intervention conditions. However, to minimize expectancy-related influences, participants were not informed of the specific hypotheses of the study. Expectation and engagement levels were documented after each session to monitor participants’ perceptions and compliance; however, these variables were not incorporated into the statistical models.

### 2.2. Procedures

This randomized crossover trial was designed to compare the acute effects of three experimental conditions—low-intensity continuous training (LICT), short-interval high-intensity interval training (SI-HIIT), and long-interval high-intensity interval training (LI-HIIT)—on cognitive performance [Trail-Making Test, Parts A and B (TMT-A, TMT-B) )] and neurochemical responses (irisin and blood lactate levels) in healthy late-adolescent males.

The participants were randomly allocated into three groups, with each group following a distinct sequence of the three experimental conditions—LICT, SI-HIIT, and LI-HIIT—according to a Latin square design. This design ensured that all conditions were equally represented across the three test sessions and that order effects were minimized by systematically rotating the sequence of conditions. A 7-day washout period was implemented between sessions to minimize any carryover effects, whether physiological or cognitive. To prevent potential residual influences, especially those related to elevated neurochemical responses such as increases in BDNF following high-intensity exercise, no two high-intensity conditions were scheduled in consecutive sessions. By the conclusion of the study, all the participants had completed each of the three conditions in a fully counterbalanced and intensity-controlled manner.

All exercise sessions were carried out under the supervision of a certified strength and conditioning coach to ensure protocol fidelity and participant compliance. Sessions took place in the morning hours (between 09:00 and 11:00) on an outdoor track and in a field facility to control for potential circadian influences. Environmental conditions were consistent across all sessions, with temperature ranging from 18 to 22 °C and relative humidity from 25 to 30%. The participants were instructed to refrain from intense physical activity for at least 48 h prior to testing, avoid alcohol and caffeine intake within the 24 h preceding each session, and maintain a minimum of 7–8 h of sleep the night before. Additionally, they were asked to keep their dietary habits and overall lifestyle consistent throughout the study period.

During the initial visit, participants’ body weight and body fat percentage were measured via bioelectrical impedance analysis (BC-418, Tanita, Tokyo, Japan). The resting heart rate (HR) was then recorded, followed by the administration of the Yo-Yo Intermittent Recovery Test Level 1 (YYIRT-1) to determine the maximal aerobic speed (MAS), aerobic capacity, and peak heart rate (HRmax).

In the subsequent three sessions, each participant completed all the experimental conditions. Each exercise session began with a standardized 10 min warm-up consisting of light jogging (at 50% MAS), dynamic mobility exercises, and preparatory movement drills. These drills included 3 × 20 m accelerations and 2 × 50 m runs with flying starts, during which participants gradually increased their speed. All exercise protocols were conducted on a standard outdoor athletics track and lasted approximately 35 min in total—comprising a 10 min warm-up and 24 min of exercise. The total session duration was selected on the basis of prior findings indicating that this length of exercise optimally supports executive function outcomes [38].

Exercise intensity and volume were individualized on the basis of each participant’s MAS derived from the YYIRT-1. Running distances were calculated via the formula time × speed, ensuring that all participants experienced equivalent training loads across conditions. Venous blood samples were collected immediately before and after each session to assess neurochemical markers. Cognitive performance was evaluated postexercise via the TMT-A and TMT-B. Throughout all exercise sessions, heart rate was continuously monitored using a Polar V800 device (Polar OY, Kempele, Finland). The Feeling Scale (FS) and ratings of perceived exertion (RPE; Borg 6–20 scale) were also recorded [39] (Figure 2).

#### Exercise Interventions

The exercise conditions are presented in Figure 3 and Table 1.

Short-interval high-intensity interval training (SI-HIIT): Participants ran at 110–120% of their MAS for 15 s intervals, covering a precalculated distance during each bout. After each 15 s sprint, they rested passively for 15 s and returned to their starting point. This process was repeated continuously for 3 min, resulting in one set. A total of four 3 min sets were completed, interspersed with 3 min passive rest periods, yielding a total exercise time of 24 min.

Long-interval high-intensity interval training (LI-HIIT): Participants ran at a speed corresponding to 80–90% of their MAS. The protocol consisted of four 4 min bouts separated by 2 min active recovery periods, during which participants ran at 50% of their MAS. This pattern totaled 24 min of exercise.

Low-intensity continuous training (LICT): Participants ran continuously for 24 min at 50–60% of their MAS without rest.

### 2.3. Measurements

#### 2.3.1. Anthropometric Assessments

A composition analyzer was used; this device applies the bioelectrical impedance technique and employs multiple frequencies (1–50 kHz) to obtain comprehensive body composition data.

The participants completed the YYIRTL-1, a progressively demanding and audio-paced test designed to evaluate aerobic capacity, in accordance with the protocol described by Bangsbo et al. [40]. Heart rate was continuously monitored during the assessment via a Polar V800 device, and the highest value obtained was considered the maximal heart rate.

#### 2.3.2. Exercise Intensity

To provide an objective assessment of exercise intensity, heart rate (HR) was continuously recorded at 5 s intervals via Polar V800 device during each session. The participants additionally wore chest straps for HR acquisition, and the average HR for each repetition period was calculated during subsequent data analysis.

#### 2.3.3. Cognitive Function Performance

Cognitive performance was evaluated via the TMT-A and TMT-B, which have been validated for Turkish young adults by Türkeş et al. [41]. This instrument assesses attention, visual scanning, visuomotor speed, and cognitive flexibility. Although the TMT-A and TMT-B are moderately correlated, the TMT-B is regarded as more sensitive to executive functioning, particularly cognitive flexibility. The assessment was administered in a paper-and-pencil format and comprised two sections. In Part A, participants were instructed to sequentially connect 25 circled numbers (1–25) in ascending order (e.g., 1–2–3…), performing the task as quickly and accurately as possible without lifting the pen. In Part B, they were required to alternate between numbers and letters in ascending alphanumeric order (e.g., 1–A–2–B… up to 13–L). Standardized instructions and a brief practice trial preceded each section to ensure comprehension of the task. The primary outcome variable for both parts was the completion time (in seconds) needed to finish the sequence correctly. Any mistakes were immediately indicated by the examiner and corrected by the participant, with the correction duration included in the total completion time.

#### 2.3.4. The Feeling Scale (FS)

Feeling was utilized to evaluate the affective response, with the following instruction: “Please choose the number that most accurately reflects your current emotions at this precise moment.” The scale ranged from −5 (very bad) to +5 (very good) [42].

#### 2.3.5. Biochemical Analysis

Venous blood samples (4 mL) were drawn from the antecubital vein of the participants in a seated position, both prior to and immediately following exercise. The samples were subsequently centrifuged at 3000× *g* for 15 min, after which the obtained serum was stored at −80 °C until further analysis. Serum irisin concentrations were quantified via an enzyme-linked immunosorbent assay (ELISA) kit following the manufacturer’s instructions (SunRed Bio Company, Cat. No: 201-12-5328; Shanghai, China). All the samples were anonymized, and identifiers were revealed only after completion of the analyses.

### 2.4. Statistical Analyses

The data are expressed as means ± standard deviations. All statistical analyses were performed using JAMOVI (Version 2.3). Changes in irisin and TMT-A and TMT-B completion times were analyzed using linear mixed-effects models with fixed effects for pre-post (time point), condition (3), time point×condition, period (1–3), and sequence (3), plus a random intercept for participants. HRmean, HRmax, HR%, and FS values were analyzed using two-way ANOVA. When necessary, Bonferroni-adjusted post hoc comparisons were applied. Effect sizes (Cohen’s d) were calculated and interpreted using the conventional thresholds (small: d = 0.1; medium: d = 0.6; large: d = 0.8) [43]. Statistical significance was set at *p* < 0.05. Data visualization was performed via JAMOVI and JASP (Version 0.19.3).

## 3. Results

In the following section of the present study, the statistical findings for LICT, LI-HIIT, and SI-HIIT are presented (Table 2 and Table 3).

Figure 4a shows that the linear mixed-effects model indicated a significant main effect of time (pre–post) on circulating irisin levels (F = 13.467, df = 1, df (res) = 48.00, *p* < 0.001). A significant main effect of condition was also observed (F = 26.455, df = 2, df (res) = 48.00, *p* < 0.001). Furthermore, the time (pre-post) × condition interaction reached significance (F = 7.236, df = 2, df (res) = 48.00, *p* = 0.002), indicating that pre–post changes in irisin differed across exercise conditions. Sequence effects were not statistically significant (F = 0.347, df = 2, df (res) = 8.00, *p* = 0.717), and period effects likewise did not reach significance (F = 2.047, df = 2, df (res) = 48.00, *p* = 0.140). The random intercept for participants was also significant (*p* < 0.001). According to the analysis, there was no significant difference in pre-exercise irisin level between the groups (*p* > 0.05). However, post-exercise comparisons revealed that the SI-HIIT group presented significantly greater increases in irisin levels than the LICT group did (*p* < 0.001, effect size = 0.81). Compared with LICT, LI-HIIT also resulted in a greater increase (*p* = 0.038, effect size = 0.77).

Figure 4b shows that the linear mixed-effects model revealed a significant main effect of time (pre–post) on TMT-A completion time (F = 22.517, df = 1, df (res) = 48.00, *p* < 0.001). A significant main effect of condition was also observed (F = 10.064, df = 2, df (res) = 48.00, *p* < 0.001). In addition, the time (pre-post) × condition interaction reached statistical significance (F = 4.796, df = 2, df (res) = 48.00, *p* = 0.013), indicating that the magnitude of pre–post changes differed among exercise conditions. Sequence effects were not statistically significant (F = 0.270, df = 2, df (res) = 8.00, *p* = 0.770), and period effects likewise did not reach significance (F = 1.262, df = 2, df(res) = 48.00, *p* = 0.292). The random intercept for participants was also significant (*p* < 0.001). According to the analysis, there was no significant difference in pre-exercise TMT-A completion time between the groups (*p* > 0.05). However, postexercise comparisons revealed that only the SI-HIIT group had a significantly shorter completion time (better performance) on the TMT-A than the LICT (*p* < 0.001, effect size = 0.71) and LI-HIIT (*p* = 0.016, effect size = 0.56) groups did.

Figure 4c shows that the linear mixed-effects model revealed a significant main effect of time (pre–post) on TMT-B performance (F = 10.133, df = 1, df (res) = 48.00, *p* = 0.003). A significant main effect of condition was also observed (F = 9.764, df = 2, df (res) = 48.00, *p* < 0.001). In addition, the time (pre-post) × condition interaction was statistically significant (F = 10.948, df = 2, df (res) =48.00, *p* <0.001), indicating that the magnitude of pre–post changes differed across exercise conditions. Sequence effects were not statistically significant (F = 1.359, df = 2, df (res) = 8.00, *p* = 0.310), and period effects likewise did not reach significance (F = 0.394, df = 2, df (res) = 48.00, *p* = 0.676). The random intercept for participants was also significant (*p* < 0.001). According to the analysis, there was no significant difference in pre-exercise TMT-B completion time between the groups (*p* > 0.05). However, postexercise comparisons revealed that only the SI-HIIT group had a significantly shorter completion time (better performance) on the TMT-B than the LICT (*p* < 0.001, effect size = 0.50) and LI-HIIT (*p* < 0.001, effect size = 0.45) groups did. 

For the feeling scale (Figure 5a), the results revealed that the group effect was significant (F = 86.83, df = 2, *p* < 0.001, $\eta_{p}^{2}$ = 0.939). According to the analysis of the feelings scale scores, the SI-HIIT group reported significantly higher scores than both the LI-HIIT (*p* < 0.001) and the LICT (*p* < 0.001) groups did. No statistically significant difference was found between the LI-HIIT and LICT groups (*p* > 0.05).

The HR mean (bpm) (Figure 5b) results revealed that the group effect was significant (F = 194.95, df = 2, *p* < 0.001, $\eta_{p}^{2}$ = 0.951). In terms of the mean HR (bpm), the SI-HIIT group presented significantly greater values than the LICT (*p* < 0.001) and LI-HIIT (*p* = 0.001) groups did. Additionally, the LI-HIIT group presented significantly greater mean HR values than did the LICT group (*p* < 0.001).

The HR% (bpm) (Figure 5c) results revealed that the group effect was significant (F = 205.16, df = 2, *p* < 0.001, $\eta_{p}^{2}$ = 0.954). In the comparison of HR% (bpm), the SI-HIIT group presented significantly higher values than both the LICT (*p* < 0.001) and LI-HIIT (*p* = 0.001) groups did. Similarly, the LI-HIIT group presented significantly greater HR% values than the LICT group did (*p* < 0.001).

The HR max. (bpm) (Figure 5d) results revealed that the group effect was significant (F = 39.64, df = 2, *p* < 0.001, $\eta_{p}^{2}$ = 0.799). The maximum HR (bpm) was significantly greater in the SI-HIIT group than in both the LICT (*p* < 0.001) and LI-HIIT (*p* = 0.010) groups. Similarly, the LI-HIIT group had significantly greater HR max values than the LICT group did (*p* < 0.001).

## 4. Discussion

The present study investigated the acute effects of short- and long-interval HIIT and LICT on circulating irisin levels and executive function in late adolescents. Both HIIT protocols elicited greater increases in irisin levels than LICT. Regarding cognitive responses, SI-HIIT produced improvements in both TMT-A and TMT-B. These findings suggest that exercise intensity and interval structure may play critical roles in the acute modulation of irisin release and executive function.

Understanding how exercise sustains brain structure and function across the lifespan is essential, yet the biological processes underlying these benefits are remarkably complex [1]. Even a single bout of exercise can trigger widespread systemic changes, altering the expression of nearly 10,000 circulating analytes, including transcripts, proteins, metabolites, and lipids, thereby underscoring the multifaceted nature of its impact on brain health [44]. Among the numerous molecules altered by exercise, myokines have emerged as key candidates linking peripheral metabolic activity to central nervous system adaptations. In this context, irisin has attracted considerable attention because of its proposed role in muscle–brain cross talk and neuroprotection [14,15,45]. Nevertheless, several studies have reported contradictory findings. Hecksteden et al. [46] reported no significant changes in irisin following aerobic endurance or strength exercise. Moreover, in a recent systematic review and meta-analysis, Cosio et al. [19] suggested that continuous endurance training elevates circulating irisin in healthy adults, whereas evidence from interval training remains limited and inconclusive. In contrast to these findings, we observed increased irisin levels following both interval-based high-intensity protocols, with greater elevations than time-matched lower-intensity continuous exercise. Findings are largely in line with previous human studies suggesting that HIIT facilitates irisin release more effectively than lower-intensity or continuous modalities do [20,21,22,23,47]. For example, Huh et al. [48] reported greater postexercise irisin levels in young adults following treadmill running at 70–75% VO_2_max until exhaustion and high-intensity swimming exercise. More recently, Léger et al. [49] demonstrated that serum irisin release is intensity dependent, exhibiting a threshold response at 50% MAS. Importantly, these findings extend this evidence by demonstrating that not only exercise intensity but also the interval structure (short vs. long intervals) critically modulates the acute irisin response.

Beyond adult findings, the limited evidence available in adolescent populations also aligns with our results, showing that HIIT elicits stronger irisin responses than continuous exercise. Colpitts et al. [21] reported that a HIIT protocol (alternating 5 min bouts at 50% and 2 min bouts at 85–90% HRreserve) elicited a markedly greater peak irisin response than continuous cycling at 50% HRreserve in healthy-weight adolescents, whereas overweight peers showed a blunted response. Similarly, Archundia-Herrera et al. [22] found that a single HIIT bout increased skeletal-muscle FNDC5/irisin protein expression in sedentary female adolescents. Similarly, data from sprint-interval swimming (two sets of 4 × 50 m maximal-intensity freestyle) in trained adolescents also demonstrate intensity- and dose-dependent effects on circulating irisin [50], further supporting the pattern observed in the present study. Moreover, Löffler et al. [23] observed acute elevations in irisin following short bouts of strenuous exercise (cycling ergometry, 15 min maximal effort) in both children and young adults, whereas habitual low-intensity physical activity did not elicit measurable changes. Collectively, these findings suggest that high-intensity, interval-structured exercise provides a sufficiently strong stimulus to acutely upregulate irisin levels during adolescence.

HIIT is known to impose larger fluctuations in metabolic and hemodynamic stress than LICT, including greater lactate accumulation and transient redox challenges, which may contribute to neurotrophic adaptations through hormetic mechanisms [9]. Animal studies have shown that such interval-induced metabolic stress can enhance neurotrophic marker expression more effectively than continuous training, partly via redox-inflammatory signaling pathways [9]. In humans, HIIT activates key metabolic regulators such as AMP-activated protein kinase (AMPK) and p38 mitogen-activated protein kinase (p38 MAPK), promoting PGC-1α–driven FNDC5 expression and subsequent irisin release [51,52]. Lactate generated during intense intervals may further stimulate the sirtuin-1 (SIRT1)–PGC-1α–FNDC5/irisin axis in the brain, supporting BDNF-related plasticity [53,54]. Collectively, these pathways may suggest that the acute metabolic and mechanical demands of HIIT more effectively trigger peripheral and central neurotrophic signaling than continuous exercise. Recent work also suggests that FNDC5/irisin expression is more pronounced in fast-twitch muscle fibers [49], which are likely recruited to a greater extent during HIIT than LICT. Although some studies have reported that overweight and obese boys and girls exhibit higher resting irisin concentrations than their normal-weight peers [55,56], individuals with lower adiposity tend to show larger postexercise irisin responses [26], which may also have contributed to the elevations observed in the present study.

In recent decades, research on the acute effects of exercise on neurocognitive performance has expanded considerably, reflecting the growing recognition of lifestyle factors as critical determinants of brain health. Although HIIT has emerged as a time-efficient modality characterized by brief bouts of intense effort interspersed with recovery, its cognitive and psychological effects remain insufficiently understood [7,57,58]. In the present study, SI-HIIT protocols significantly improved performance (shorter completion times) on the TMT-A and B than LICT and LI-HIIT. Consistent with these findings, prior research has also indicated that acute bouts of HIIT can enhance cognitive function in young adults [59,60]. Kao et al. [61] reported that low-volume HIIT was more effective than continuous aerobic exercise in enhancing inhibitory control, as evidenced by EEG measures of faster processing speed and reduced attentional resource allocation. In a randomized trial with young adults, Mekari et al. [60] demonstrated that, compared with MICT, HIIT produced superior improvements in executive function (Trail-Making Test B and Stroop switching task), suggesting that interval structures may be more effective for enhancing higher-order cognitive processes. More recently, studies by Yue et al. [62] and Mou et al. [63] have provided further support, demonstrating that acute HIIT protocols can yield superior or more sustained cognitive benefits relative to continuous exercise, particularly in young adults. In contrast to these findings, a recent meta-analysis suggested that HIIT and MICT are equally effective in enhancing inhibitory control among healthy adults, with no significant differences between modalities [64]. In addition, our previous study in young soccer players highlighted that cognitive improvements did not occur following a running-based HIIT protocol, whereas game-based HIIT specifically enhanced performance only on the TMT-A [65].

Evidence from adolescent populations also indicates that acute exercise intensity may play a meaningful role in shaping executive performance. Recent meta-analytic findings further show that engaging in at least moderate-intensity physical activity supports cognitive development in adolescents [66], underscoring the relevance of exercise intensity for this age group. Moreau et al. [67] showed that cognitive benefits are not limited to traditional aerobic exercise; HIIT formats can represent a practical and effective alternative to longer-duration workouts for eliciting cognitive improvements in pre-adolescents. A study by Hatch et al. [68] reported that a 30 min running-based HIIT session elicited greater post-exercise cognitive improvements in adolescents than a 60 min session. This finding suggests that shorter, time-efficient, high-intensity protocols may be particularly effective in this age group, which may also help explain the cognitive gains observed in the present study using a 24 min HIIT format. A recent systematic review by Reyes-Amigo et al. [69] reported that although active exercise breaks during school hours may positively influence classroom behavior in children and adolescents, their effects on executive functions remain inconclusive. This highlights the need for more targeted and effective exercise strategies. Incorporating HIIT formats into school-based programs may offer a more promising approach for specifically enhancing executive function. Similarly, several studies have reported that HIIT modalities implemented in school settings or in extracurricular contexts have the potential to improve executive function and mental health in adolescents [70,71].

Myokines are muscle-derived cytokines released during exercise that act via endocrine, paracrine, and autocrine pathways to modulate metabolism in various organs, including the brain [72]. Among them, irisin has been widely proposed as a key mediator of muscle–brain communication. Experimental and human studies indicate that irisin can enhance BDNF expression, which plays a central role in synaptic plasticity, learning, and memory [49,53,73,74]. Irisin has also been linked to reductions in neuroinflammation and improved cellular resilience, suggesting a broader neuroprotective profile. Additionally, intermittent high-intensity exercise elicits greater increases in prefrontal cortical blood flow and oxygenation than continuous exercise [58], providing a hemodynamic pathway that may complement irisin-associated molecular signaling. Although the precise contribution of the irisin–BDNF axis to cognitive enhancement remains to be fully clarified, these multimodal pathways offer a plausible explanation for the superior improvements in executive function observed following HIIT in the present study. This interpretation is consistent with prior evidence showing that HIIT can induce acute neurotrophic and metabolic responses that facilitate executive performance [75,76]. Nevertheless, the optimal exercise modality and intensity for maximizing acute cognitive benefits remain uncertain, underscoring the need for further mechanistic research [64].

This study has several limitations. First, the sample was limited to late-adolescent males, which restricts the extrapolation of findings to other populations, such as females, older adults, or individuals with clinical conditions. Second, the intervention was confined to running-based protocols. The absence of other exercise modalities, such as resistance, sport specific, or multimodal training, may produce distinct physiological and cognitive effects [77]. Third, irisin was measured at only a single postexercise time point, which may not adequately capture its dynamic secretion profile. Prior evidence indicates that irisin levels peak at different time points depending on exercise intensity and participant characteristics [78].

Future research should address current limitations by including more diverse populations (e.g., females, older adults, or clinical groups) and incorporating different exercise modalities beyond running-based protocols. Longitudinal approaches may help clarify whether acute irisin and cognitive responses translate into sustained adaptations. Moreover, integrating multiple postexercise blood collections with advanced neurobiological markers (e.g., IGF-1, VEGF, cortisol) and multimodal neuroimaging tools would enable a richer characterization of the pathways linking exercise to cognitive outcomes. From an applied perspective, although HIIT is often perceived as demanding, its superior acute biochemical and cognitive effects, together with participants’ reported preferences, highlight its feasibility as a practical, nonpharmacological strategy for supporting brain health in early adulthood and beyond.

## 5. Conclusions

This study demonstrated that both short- and long-interval HIIT protocols elicited greater acute increases in circulating irisin and SI-HIIT, further conferring unique benefits on executive function. These findings highlight the importance of exercise intensity and interval design in modulating neurochemical and cognitive responses. While further research is needed to confirm these effects across broader populations and long-term effects, results suggest that HIIT represents a practical, non-pharmacological strategy to support brain health in late adolescents. In light of these findings, it is recommended that policymakers and practitioners consider integrating HIIT protocols into youth physical activity programs to support early-life brain health.

## Figures and Tables

**Figure 1 healthcare-13-03242-f001:**
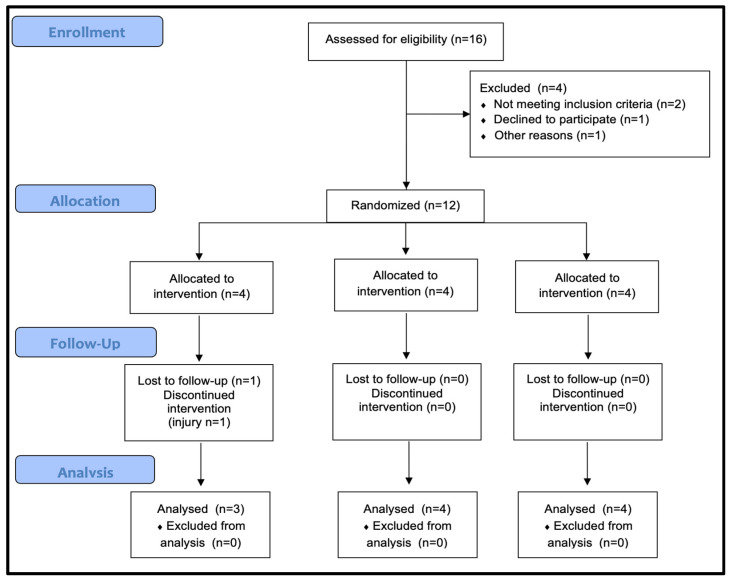
Consolidated Standards of Reporting Trials (CONSORT) flowchart.

**Figure 2 healthcare-13-03242-f002:**
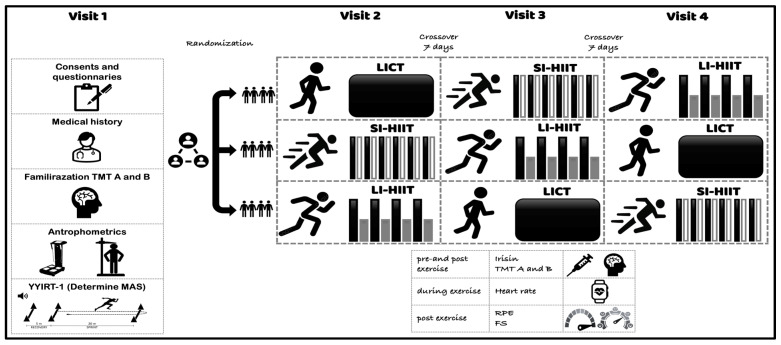
Study design. SI-HIIT: Short-interval high-intensity interval training; LI-HIIT: Long-interval high-intensity interval training; LICT: Low-intensity continuous training; YYIRT-1: Yo-Yo Intermittent Recovery Level 1 Test; MAS: maximum aerobic speed; TMT: Trail-Making Test. FS: Feeling Scale. RPE: rating of perceived exertion.

**Figure 3 healthcare-13-03242-f003:**
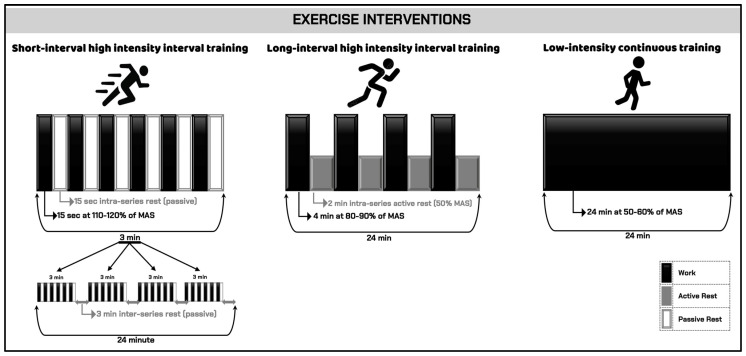
Exercise interventions. MAS: maximum aerobic speed.

**Figure 4 healthcare-13-03242-f004:**
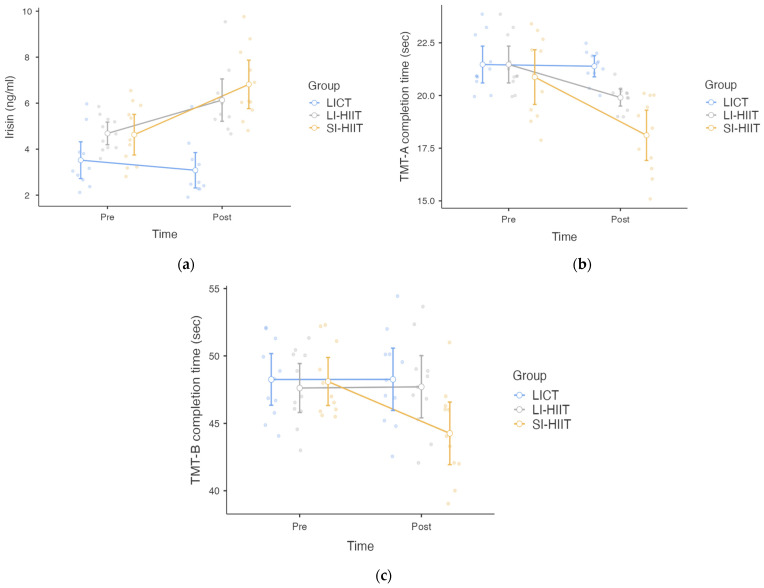
Figure presents the serum irisin responses and Trail-Making Test, Parts A and B (TMT-A and TMT-B) completion time for the low-intensity continuous training (LICT), long-interval high-intensity interval training (LI-HIIT), and short-interval high-intensity interval training (SI-HIIT) groups. (**a**) Serum irisin levels measured pre- and post-exercise are shown; (**b**) TMT-A completion time measured pre- and post-exercise are shown. (**c**) TMT-B completion time measured pre- and post-exercise are shown.

**Figure 5 healthcare-13-03242-f005:**
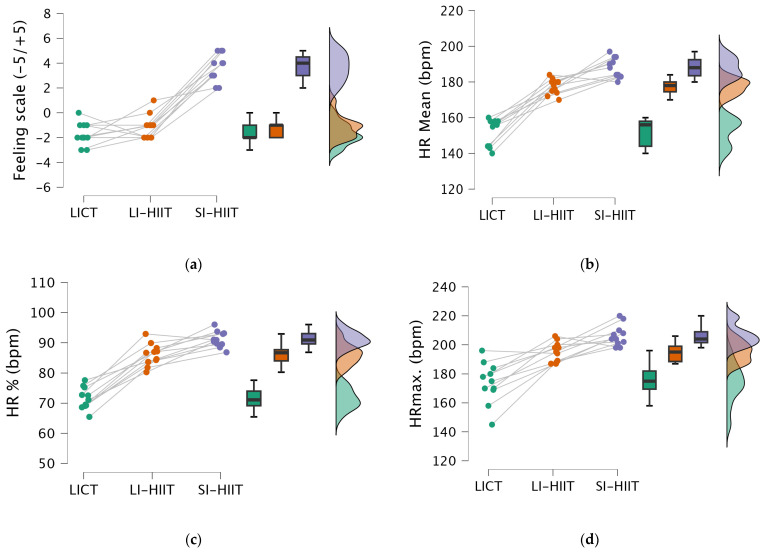
Figure presents the feeling scale (FS) scores, mean heart rate (HRmean), HR% values and maximum HR (HRmax) for the low-intensity continuous training (LICT), long-interval high-intensity interval training (LI-HIIT), and short-interval high-intensity interval training (SI-HIIT) groups. (**a**) FS scores measured postexercise are shown; (**b**) HR mean values measured during exercise are shown. (**c**) HR% values measured during exercise are shown. (**d**) HR max values measured during exercise are shown. bpm: beats per minute.

**Table 1 healthcare-13-03242-t001:** Characteristics of the exercise interventions.

Session	No. × Duration of Bout	Work Velocity	No. × Duration of Series	Intraseries Recovery	Interseries Recovery	Total Duration
SI-HIIT	6 × 15 s	110–120% MAS	4 × 3 min	15 s passive	3 min passive	24 min
LI-HIIT	1 × 4 min	80–90% MAS	4 × 4 min	–	2 min active (50% MAS)	24 min
LICT	1 × 24 min	50–60% MAS	1 × 24 min	–	–	24 min

LICT: low-intensity continuous training; SI-HIIT: short-interval high-intensity interval training; LI-HIIT: long-interval high-intensity interval training. MAS: maximum aerobic speed.

**Table 2 healthcare-13-03242-t002:** Irisin, Trail-Making Test, Parts A and B (TMT-A and TMT-B), and responses to experimental conditions.

	LICT		LI-HIIT		SI-HIIT	
	Pre	Post	Pre	Post	Pre	Post
	Mean ± SD	Mean ± SD	Mean ± SD	Mean ± SD	Mean ± SD	Mean ± SD
Irisin (ng/mL)	3.52 ± 1.19	3.08 ± 1.15	4.69 ± 0.73	6.13 ± 1.37	4.63 ± 1.31	6.82 ± 1.57
TMT-A completion time (s)	21.47 ± 1.30	20.84 ± 0.76	20.85 ± 1.36	19.90 ± 0.60	20.87 ± 1.93	18.11 ± 1.77
TMT-B completion time (s)	48.26 ± 2.84	48.27 ± 3.43	47.63 ± 2.70	47.72 ± 3.43	48.11 ± 2.65	44.26 ± 3.46

LICT: Low-intensity continuous training; LI-HIIT: Long-interval high-intensity interval training; SI-HIIT: Short-interval high-intensity interval training; ng/mL: nanograms per milliliter; s: second.

**Table 3 healthcare-13-03242-t003:** Feeling scale and heart rate responses to experimental conditions.

	LICT	LI-HIIT	SI-HIIT
	Mean ± SD	Mean ± SD	Mean ± SD
Feeling Scale (score)	−1.73 ± 0.90	−1.10 ± 0.94	3.63 ± 1.12
HRmean (bpm)	147.46 ± 5.20	177.36 ± 4.34	188 ± 5.59
HR%	71.54 ± 3.63	86.04 ± 3.62	91.15 ± 2.63
HRmax (bpm)	173.91 ± 14.10	195.10 ± 6.77	206.36 ± 7.32

LICT: Low-intensity continuous training; LI-HIIT: Long-interval high-intensity interval training; SI-HIIT: Short-interval high-intensity interval training; HR: Heart rate; HRmax: Maximum heart rate; bpm: Beats per minute.

## Data Availability

The data are available on reasonable request from the Y.Z.B., subject to ethical and privacy considerations to protect participant identity.

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
