# Peer review of "Comparison of Acute Irisin and Cognitive Responses to Different Exercise Modalities Among Late Adolescents"

_healthcare, 2025, doi:10.3390/healthcare13243242_

Round 1

Reviewer 1 Report

Comments and Suggestions for Authors

Introduction

  1. Please avoid general statements such as “Exercise is medicine” unless followed by specific evidence.
  2. Please clarify conceptual gap, that is, why late-adolescents as your study population? Provide rationale for focusing on this age.

Methods

  1. The sample size justification is detailed but unclear; specify which variable was used for effect size derivation and include reference justification.
  2. Eleven participants completed all sessions—explicitly state whether any dropout occurred.
  3. Exclusion criteria are extensive; please consider simplifying to maintain readability.
  4. Please define all abbreviations at first mention in tables (LICT, LI-HIIT, etc.).
  5. Please specify environmental conditions (temperature, humidity), given outdoor testing.

Discussion

  1. In this section, please begin with a summary by restating the main findings clearly in one short paragraph.
  2. Please simplify mechanistic pathways; current section is dense with biochemistry that distracts from empirical findings.
  3. Please compare findings to at least two adolescent studies (not adult-only).
  4. Please Add practical implications for school-based or youth fitness settings.

Reviewer 2 Report

Comments and Suggestions for Authors

In my ratings above, I included statistical design under "research design". Their underlying research design was fine.

The introduction was very well written and informative. I would suggest the authors add a brief mention of the general literature on interval length for HIIT training. For example, the impact on V02max. Readers should be able to balance all of the health effects of HIIT training protocols.

It is not clear whether and how expectations and engagement levels were “statistically controlled during the analyses”.  Please specify exactly where those variables entered the analysis.

If you planned on 16 participant total in the study, why stop at 12? The 4 who were excluded typically wouldn’t be counted toward recruitment goals.

On page 5 of 18, line 203 I think you meant “Throughout” rather than “Without”.

The statistical model is misspecified. Given the 3×3 Latin-square crossover with pre/post measures, the appropriate primary analysis is a linear mixed-effects model with fixed effects for timepoint (pre/post), condition (3), timepoint×condition, period (1–3), and sequence (3), plus a random intercept for participant. This allows estimation of the timepoint×condition interaction while adjusting for period/sequence and testing carryover. 

Given the authors are not conducting a fully counterbalanced Latin square design (i.e., 3! sequences), it is critical that sequence is a fixed effect in the model for reasons considered foundational in the research design literature.

Although I agree with the authors that carryover effects are unlikely, they should be formally tested.

In Table 2, there were notable differences at baseline on irisin. Were these tested as paired differences? What, precisely, was the statistical model used to compare baseline measurements? Please report paired baseline comparisons across conditions.

Round 2

Reviewer 1 Report

Comments and Suggestions for Authors

Comments addressed

Comments on the Quality of English Language

Acceptable

Author Response

We would like to express our gratitude for your thoughtful comments and for indicating that all concerns have been satisfactorily addressed.

Thank you for your time, effort, and contribution to the improvement of our manuscript.

Best regards. 

Reviewer 2 Report

Comments and Suggestions for Authors

The paper is much improved. My only remaining concern is that, although the authors tested for carry over effects, the model remains misspecified.

As I noted in the previous review, the appropriate primary analysis is a linear mixed-effects model with fixed effects for timepoint (pre/post), condition (3), timepoint×condition, period (1–3), and sequence (3), plus a random intercept for participant

The current model is described as: "linear mixed-effects models with fixed effects for condition, time, and their interaction." I assume the model includes timepoint (unless you just looked as posttests) and you ignore period and sequence effects. I also assume you included a random effect for participant.

Your statistical consultant should be able to guide you in the set up of this model. Your power analysis should refer to the correctly specified model.
